# Redox reactions and weak buffering capacity lead to acidification in the Chesapeake Bay

Wei-Jun Cai [1], Wei-Jen Huang[1,2], George W. LutherIII [1], Denis Pierrot[3], Ming Li[4], Jeremy Testa[5], Ming Xue[1,6], Andrew Joesoef[1], Roger Mann[7], Jean Brodeur[1], Yuan-Yuan Xu[1], Baoshan Chen [1], Najid Hussain[1], George G. Waldbusser [8], Jeffrey Cornwell[4] & W. Michael Kemp[4]

The combined effects of anthropogenic and biological $CO_2$ inputs may lead to more rapid acidification in coastal waters compared to the open ocean. It is less clear, however, how redox reactions would contribute to acidification. Here we report estuarine acidification dynamics based on oxygen, hydrogen sulfide ($H_2S$), pH, dissolved inorganic carbon and total alkalinity data from the Chesapeake Bay, where anthropogenic nutrient inputs have led to eutrophication, hypoxia and anoxia, and low pH. We show that a pH minimum occurs in mid-depths where acids are generated as a result of $H_2S$ oxidation in waters mixed upward from the anoxic depths. Our analyses also suggest a large synergistic effect from river–ocean mixing, global and local atmospheric $CO_2$ uptake, and $CO_2$ and acid production from respiration and other redox reactions. Together they lead to a poor acid buffering capacity, severe acidification and increased carbonate mineral dissolution in the USA's largest estuary.

[1] School of Marine Science and Policy, University of Delaware, Newark, DE 19716, USA. [2] Department of Oceanography, National Sun Yat-sen University, Kaohsiung 80424, Taiwan. [3] RSMAS, University of Miami, Miami, FL 33149, USA. [4] Horn Point Laboratory, University of Maryland Center for Environmental Science, Cambridge, MD 21613, USA. [5] Chesapeake Biological Laboratory, University of Maryland Center for Environmental Science, Solomons, MD 20688, USA. [6] State Key Laboratory of Petroleum Pollution Control, CNPC Research Institute of Safety and Environmental Technology, Beijing 102206, China. [7] Virginia Institute of Marine Science, Gloucester Point, VA 23062, USA. [8] College of Earth, Ocean, and Atmospheric Sciences, Oregon State University, 104 COAS Admin. Bldg., Corvallis, OR 97331, USA. Correspondence and requests for materials should be addressed to W.-J.C. (email: wcai@udel.edu)

Anthropogenic carbon dioxide ($CO_2$) has increased more rapidly in the atmosphere since the Industrial Revolution than natural $CO_2$ increase in any period of the last ~800,000 years[1, 2]; consequently, it has been known that the uptake of $CO_2$ by the ocean has altered surface seawater acid-based chemistry lowering pH by about 0.1 unit and calcium carbonate saturation state by roughly 0.5. This process, known popularly as ocean acidification (OA) for over a decade, will continue to decrease seawater pH by about 0.3 units by the end of the century[3, 4]. It is likely that OA will cause detrimental effects on the health of marine organisms and ecosystems and alter the associated biogeochemical processes[5–7].

Recent research indicates that eutrophication can exacerbate OA, where respiratory processes contribute a far greater acidification in the coastal oceans relative to the open ocean[8–13]. Coastal eutrophication occurs with increased inputs of nutrients from the application of chemical fertilizers, discharges of human and animal wastes, and atmospheric $NO_x$ inputs from fossil fuel burning, which have fueled large algal blooms in many coastal water bodies, especially those near population centers[14]. It is well known that decomposition of algal organic matter from highly productive surface water leads to the development of seasonally low oxygen (hypoxic) or even zero oxygen (anoxic) bottom waters in many coastal water bodies in the world[15, 16]. However the coupling between redox and acid–base chemistry has not been explored extensively in seasonally anoxic and partially mixed estuaries nor in permanently anoxic deep basins although redox chemistry and pH have been reported before in the latter[13, 17–22]. Specifically, it is not known how subsurface water pH dynamics are influenced by anaerobic respiration and the oxidation of reduced chemical species (notably $H_2S$) in seasonally low oxygen ($O_2$) estuaries around the world let alone the interaction of these processes with the anthropogenic $CO_2$ induced OA.

The Chesapeake Bay is the largest estuary in the United States with a well-documented history of eutrophication over the past half century[23–25]. A recent report demonstrates that some regions of the bay have suffered a long-term pH decline related to eutrophication[26]. However, few process studies have examined the $CO_2$ system and pH in the Chesapeake, and those that exist have focused on tributaries in the southern reaches of the estuary[27, 28]. To address the coupling between acid–base chemistry and redox chemistry and its contribution to coastal OA, we sampled the water column repeatedly for several days within a deep basin of the main-stem bay in August 2013 and 2014, a time of peak hypoxia and anoxia, and these data were supplemented with an April 2015 (pre-hypoxia) study. In this paper, we report and explain the occurrence of a pH minimum at and above the oxic–anoxic boundary due to $H_2S$ oxidation. We further demonstrate how a combination of processes drives down pH and aragonite mineral saturation state, leading to $CaCO_3$ mineral dissolution in subsurface waters. Finally, we present a general geochemical model to explain why large eutrophic estuaries, exemplified by the Chesapeake Bay, are particularly vulnerable to the acidification stresses caused by the increase of anthropogenic atmospheric $CO_2$ and aquatic eutrophication and respiration.

## Results

**Acidification due to eutrophication-induced local $CO_2$ uptake.** Partial pressure of carbon dioxide ($pCO_2$) in surface waters of the Chesapeake Bay exceeded 1600 µatm in its upper reach and was below the atmospheric level (~ 390 µatm) in the mid-bay for all three cruises between spring and summer (Fig. 1a, b, Table 1, Supplementary Fig. 1, and Methods). The low $pCO_2$ was accompanied by high chlorophyll-a, a phytoplankton biomass proxy, and supersaturated dissolved $O_2$ for much of the year in the mid- and lower-bay (Supplementary Figs. 2–5), indicating net biological production fueled by high riverine nutrient loading. The low surface $pCO_2$ should lead to atmospheric $CO_2$ invasion

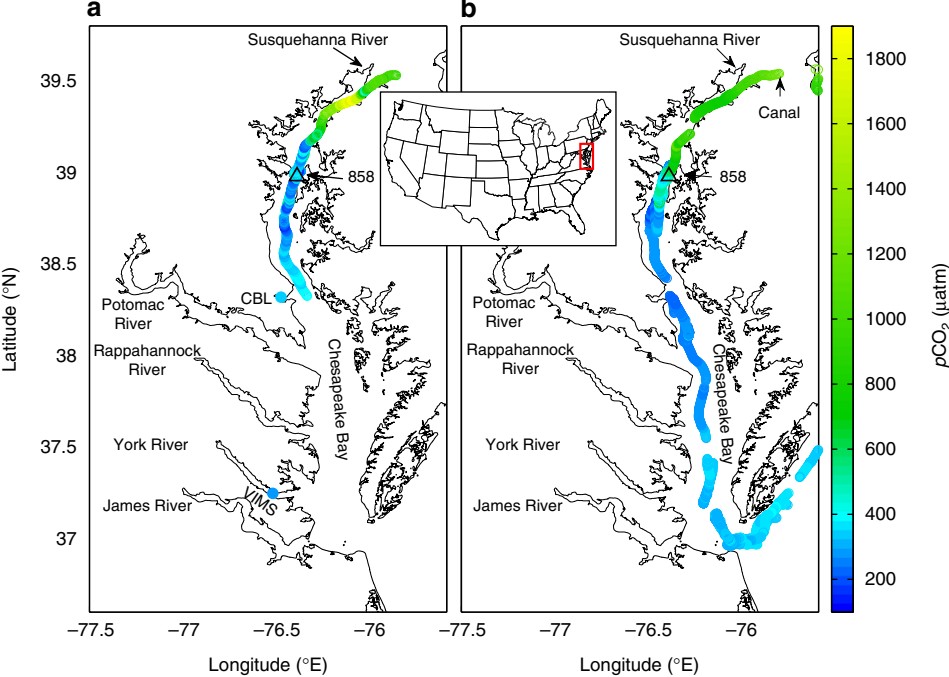

**Fig. 1** Measured surface water $pCO_2$ overlapped on a site map of the Chesapeake Bay. **a** 8–15 August 2013 and **b** 11–16 April 2015. Data were collected underway aboard R/V Sharp and supplemented with dockside measurements at the Chesapeake Biological Laboratory (CBL) and Virginia Institute of Marine Science (VIMS) during the August cruise. The inserted map shows the general location of the bay. Station 858 is our focused study site. The cruise average of atmospheric and surface water $pCO_2$ values and monthly wind speed are given in Table 1. The $pCO_2$ range was 340–590 µatm near CBL and 290–550 µatm near VIMS, with the lower ends represent incoming bay water during high tides and the high ends representing outgoing sub-estuarine waters during low tides

and may contribute to water column $CO_2$ accumulation and acidification, particularly given atmospheric concentrations are ~40% greater than the pre-industrial and bay has a long water residence time of 100 days[24]. While complete water column mixing and destratification occurs occasionally during storms[29, 30], smaller wind events more frequently mix water and chemical species down to middle depths (Fig. 2a, Supplementary Figs. 6 and 7). Turbulence in the tidally driven bottom boundary layer will then mix the chemical species in the bottom water[31]. We have estimated the air-to-water $CO_2$ flux and its impact on water column total dissolved inorganic carbon (DIC) and pH over the period of spring to summer (Table 1). To calculate the effect of DIC increase on bottom water pH decrease, we have modified the popularly used CO2SYS program to include

$H_2S$–$HS^-$ and $NH_3$–$NH_4^+$ species in the acid–base equilibrium calculations as the bottom water in August contains these reduced chemical species (Methods). The resulting bottom-water pH decreases (0.08–0.13 over the spring-summer period; Table 1) are significant when compared with pH decrease due to $CO_2$ uptake from the atmosphere in the open ocean (0.11). However the time scales of acidification due to local $CO_2$ uptake (months) are much shorter than open ocean uptake (decadal to centennial).

We also note that acidification induced by local $CO_2$ uptake is caused by both increased atmospheric $CO_2$ and coastal eutrophication. This is in sharp contrast with the $CO_2$ uptake in the open ocean where atmospheric forcing is comparable to the coastal ocean but biological $CO_2$ removal and physical mixing are less intense or frequent. Clearly, climate change, anthropogenic

| Table 1 Air–sea $CO_2$ flux and its impact on bottom-water DIC and pH in the middle Chesapeake Bay | | | | | | |
|---|---|---|---|---|---|---|
| Month Year | Air $pCO_2$ | Water $pCO_2$ | Wind speed | $CO_2$ flux | ΔDIC | ΔpH |
| | µatm | µatm | m s$^{-1}$ | mmol m$^{-2}$ d$^{-1}$ | µmol kg$^{-1}$ | |
| August 2013 | 380.2 ± 11.2 | 309.1 ± 99.3 | 4.4 ± 2.3 | −4.3 ± 0.4 | 21.2 | −0.081 |
| August 2014 | 373.1 ± 9.1 | 251.6 ± 70.9 | 4.2 ± 2.2 | −6.7 ± 0.7 | 33.1 | −0.13 |
| April 2015 | 409.1 ± 8.8 | 341.3 ± 116.6 | 5.8 ± 3.6 | −6.6 ± 0.9 | 32.6 | – |

The mid-bay region is defined as the area between 37.9 and 39.0° N. Monthly averaged wind data were calculated from National Data Buoy Center station# COVM2-8577018 at Cove Point LNG Pier, MD

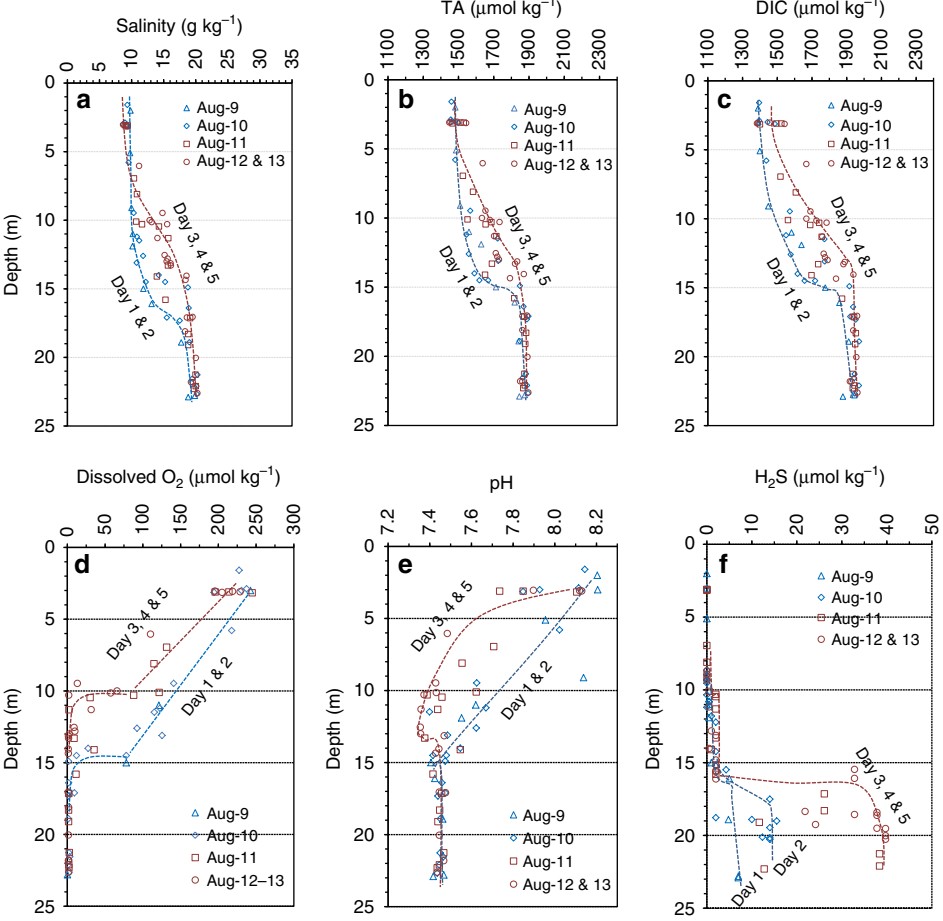

Fig. 2 Vertical distributions of measured chemical properties at the focused study site in August 2013. a Salinity, b total alkalinity (TA), c dissolved inorganic carbon (DIC), d dissolved oxygen, e pH (25 °C and NBS scale), and f $H_2S$ concentration. See Fig. 1 for location (station 858). The lines are the lower and upper boundaries between days 1 and 2 and days 3–5, respectively. August 2014 data are presented in Supplementary Fig. 8 for comparison and affirmation of the 2013 observations

inputs, and natural processes have jointly altered the carbon cycle and stressed aquatic environments in the coastal zone.

The spatial gradients of $pCO_2$ observed here and inferred net autotrophy are consistent with prior investigations using oxygen-based approaches to measuring primary production and respiration. Kemp et al[25] concluded that the Chesapeake Bay was net autotrophic overall, but heterotrophic conditions (where respiration exceeded photosynthesis) prevailed in low-salinity regions where we measured supersaturated $pCO_2$ (Fig. 1a, b, Supplementary Fig. 1) and $O_2$ was consistently under saturated (Supplementary Figs. 2b, 3–5). Dissolved $O_2$ tends to be undersaturated in northern regions of the bay given high respiration rates associated with external loads of organic carbon[32].

These heterotrophic conditions gave way to a near balanced and an autotrophic metabolism in the mid- and lower-bay, leading to a mean, bay-wide net ecosystem production, which is consistent with strong under saturation of $pCO_2$ in these seaward regions (Fig. 1a, b and Supplementary Fig. 1). The mid- and lower-bay stations (CB3.3C and south) tended towards $O_2$ supersaturation during most months of the year, especially during the warmer months (Supplementary Figs. 2b, 4 and 5). Despite some interannual variability in the seasonal pattern of dissolved $O_2$ saturation, the years of 2013–2015 indicate similar seasonal patterns. Oxygen-based estimates of metabolism showed consistent surface-layer net $O_2$ production and bottom-layer net $O_2$ consumption, the rates of which were highly correlated[32]. Although oxygen-based methods could not be applied under oxygen-depleted conditions, independent measures of sulfate reduction (SR) in sediments, which dominated the benthic metabolism during warm months and led to significant sediment–water sulfide fluxes in the mid-bay[33, 34], clearly support the accumulation of sulfide observed in August 2013 (Fig. 2f).

**Subsurface pH minimum due to oxidation of reduced chemicals**. Repeated vertical profiles during both summers revealed a consistent pH minimum below the surface mixed layer at our focused study station, a deep site in the upper part of the mid-bay. Salinity profiles at this site (Fig. 2a) combined with a time series of wind speed indicate a physical mixing event before our first sampling on August 9, 2013. Stratification quickly re-established when wind speed reduced and the wind direction switched from favoring mixing to favoring stratification (Supplementary Fig. 6a, b). Total alkalinity (TA) and DIC were lower in the surface, but became higher in the bottom water (Fig. 2b, c). Dissolved $O_2$ was at saturation or supersaturation in the surface due to gas exchange and biological production and was not detectable below 10–15 m depths due to respiration (Fig. 2d).

On day 1, the mixed layer depth was still as deep as 15–18 m, but within 2 days, it shoaled to 10 m (Fig. 2a). Following this dynamic change, the $O_2$ penetration depth changed from about 15 m on day 1 to about 10 m on days 3–5 (Fig. 2d). Simultaneously, water column pH (25 °C and NBS scale) decreased greatly over this period (Fig. 2e). For example, at the depth of 6 m, pH decreased from nearly 8.0 on day 1 to ~ 7.5 on days 3–5. A pH minimum (7.35 ± 0.03) occurred at 11–13 m depth in the low $O_2$ zone (<10 µmol kg$^{-1}$), below which pH increased slightly and then became constant at 7.45 ± 0.02. This pH minimum and the associated rapid pH decrease above it within a short period of <2 days have not been previously documented, although large pH changes were observed or expected in many strongly productive or stratified shallow water systems[35–37]. Such dramatic decreases in both $O_2$ and pH over just 2 days could likely put the biological system under stress[38]. This pH minimum was also observed in August 2014, though water column $O_2$ and pH profiles in August 2014 were relatively

stable before and during the 5-days cruise, as wind speeds were less strong and less variable than those of the 2013 (Supplementary Fig. 7a, b), the oxycline (where $O_2$ decreases rapidly) and the pH minimum were even sharper and shallower, and bottom-water pH, DIC and TA were lower in 2014 than 2013 (Supplementary Fig. 8).

We suggest the oxidation of reduced chemicals is responsible for the pH minimum in the low $O_2$ zone and the rapid pH decrease above it where declining $O_2$ gradients were steepest (Fig. 2e). The coupling between acid–base and redox chemistry is described by the following formula:

$$H_2S + 2O_2 = 2H^+ + SO_4^{2-} \tag{1}$$

$$NH_4^+ + 2O_2 = 2H^+ + NO_3^- + H_2O \tag{2}$$

$$Mn^{2+} + 0.5O_2 + H_2O = 2H^+ + MnO_2 \tag{3}$$

$$Fe^{2+} + 0.25O_2 + 1.5H_2O = 2H^+ + FeOOH \tag{4}$$

In August 2013, because of the strong mixing event prior to our cruise, the total concentration of $H_2S$ was only 5 µmol kg$^{-1}$ at the 20 m depth on day 1, but it rapidly increased to 30–40 µmol kg$^{-1}$ on days 3–5 when the water column was restratified (Fig. 2f)[39]. Oxidation of other reduced chemicals accumulating in the bottom water could also have contributed to the formation of the pH minimum. $NH_4^+$ concentration measured near our site was 15–20 µmol kg$^{-1}$ at 20 m (Supplementary Fig. 9). Also, during days 3–5, bottom water [$Mn^{2+}$] and [$Fe^{2+}$] became as high as 7 and 2 µmol kg$^{-1}$, respectively[39]. When these reduced species (total concentration ~ 60 µmol kg$^{-1}$) were mixed upward into oxygenated water, they were oxidized, hydrogen ions were generated, TA was decreased and thus the water became more acidified (see Eqs. (1)–(4)). However, we recognize the oxidation of reduced species are often complex involving many intermediate steps and side products[40] and could have different $H^+$ production ratios.

It has been shown that oxidation of $H_2S$ by $O_2$ is sufficiently slow that $H_2S$ can be brought near to the surface during vigorous mixing events and lead to fish kills in coastal waters[20, 36, 41]. Similarly, ammonia oxidation is not instantaneous[12, 17, 41]. We suggest that the slow oxidation kinetics and rapid mixing facilitate the transport of reduced species and can subsequently result in acidification of the oxygenated near- and sub-surface waters, potentially resulting in a negative impact on aquatic organisms[38]. If, for example, one volume of bottom water of 60 µmol kg$^{-1}$ of reduced chemicals is mixed with one volume of sufficiently oxygenated water, the resulting mixed water has the total concentration of the reduced chemicals halved to 30 µmol kg$^{-1}$, and eventually ~ 60 µmol kg$^{-1}$ of acid (or −ΔTA) would be generated due to the oxidation of the reduced chemical species (Eqs. (1)–(4)). Based on CO2SYS simulations, the predicted pH decrease due to these oxidation reactions can be up to 0.20 pH units in Chesapeake Bay waters, although other mixing ratios and incomplete reactions due to slow kinetics may generate less of a pH decrease (Methods). This pH decrease is substantial and is consistent with our observations (Fig. 2e; also see a model simulation of TA, DIC, $O_2$, and pH evolving loci below). Note that while the size and location of this pH minimum may vary depending on the strength of the physical mixing and [$H_2S$] in the bottom water, it occurs whenever bottom-water anoxia exists regardless of whether a prior severe mixing event has occurred as in our 2013 study, because moderate mixing occurs constantly in the bay (Supplementary Fig. 7).

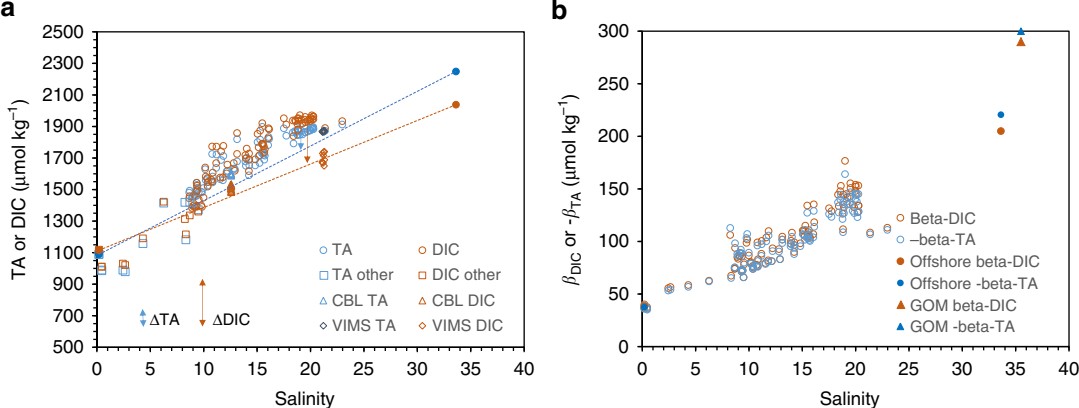

**Fig. 3** Total alkalinity and dissolved inorganic carbon as well as buffer factor distributions against salinity in August 2013. **a** Total alkalinity (*TA*) and dissolved inorganic carbon (*DIC*) and **b** buffer factors (see definitions in Eqs. (6) and (7)). Most data were collected at the focused study site over a 5-day period, as marked in Fig. 1 (station 858). Those marked as TA other or DIC other were collected in waters between station 858 and the Chesapeake–Delaware Canal in the upper bay (Fig. 1). Data were also collected in the mid- and lower-bays at the CBL dock and the VIMS dock shortly after the main cruise

In sediment porewater, a pH minimum was reported at and above the $O_2$ penetration depths as a result of oxidation of reduced chemicals, which diffused upward from deeper, anoxic depths[42, 43] and was predicted by sediment diagenetic models[44, 45]. Such a pH minimum was also seen in low $O_2$ waters of permanently stratified and anoxic deep basins including the Baltic Sea[13, 17], the Black Sea[18, 19], the Framvaren Fjord[21], the Hunnbunn Fjord[20], and the Cariaco Basin[22] though no one has pointed out this phenomenon except Yao and Millero[21] who commented that "the low pH is difficult to explain". The pH minimum is an interesting feature that results from the decrease in TA:DIC ratio due to acid production during oxidation of reduced chemicals when encountering free $O_2$ due to vigorous physical mixing. To our knowledge, this is the first time that such a pH minimum has been reported and properly interpreted in the water column. We predict that the pH minimum should occur in all oceanic systems that have seasonally or permanently occurring oxic–anoxic boundaries, including the above mentioned cases as well as in the dead-end canals of Delaware Inland Bays[36], Lake Grevelingen (the Netherlands)[12], the Saanich Inlet[46], and estuaries and bays elsewhere[17, 20]. We further argue that the pH minimum is likely more dynamic in seasonally anoxic coastal systems than permanently anoxic deep basins, due to the shallower water depth and higher frequency of physical disturbances. Physical disturbances such as winds and tides occur regularly in the Chesapeake Bay (Supplementary Fig. 7a, b) and other seasonally stratified coastal waters[12, 36]. Therefore, their chemical and biological consequences, in the context of coastal OA and deoxygenation, deserve further attention.

**Geochemical drivers and carbonate dissolution**. To separate biological processes from physical mixing and to explore the biogeochemical control mechanisms in a broader context, we examine TA and DIC vs. salinity relationships at this site together with data from other areas of the bay and the river and offshore endmembers (Fig. 3a). Between the river and ocean endmembers, as expected, TA and DIC increased with salinity. However, at our focused study station, all subsurface and bottom-water samples were located well above the mixing lines, indicating net release of $CO_2$ and accumulation of DIC and TA. In addition, both DIC and TA data collected at the Chesapeake Biological Laboratory (CBL) dock, downstream of our focused study site at the lower end of the mid-bay, were also above the mixing lines. Those from the Virginia Institute of Marine Science (VIMS) dock, farther

downstream near the bay mouth, however, showed the least enrichment relative to the conservative mixing lines. We also calculated the acid–base buffer factors from TA, DIC, and nutrients ($PO_4^{3-}$, $H_2S$, and $NH_4^+$) (Methods). It is clear the bay waters are poorly buffered as indicated by their much lower buffer factors compared to offshore waters here and elsewhere (Fig. 3b; also see next section for definitions and explanations).

TA is usually a good conservative tracer of river–ocean mixing within an estuary because it is not influenced by $CO_2$ addition and removal. Because TA and DIC share a common major component ($HCO_3^-$), deviations of DIC from the nearly conservative behavior defined by TA and salinity provide a measure of biological use or release of $CO_2$[47]. Bottom waters in the Chesapeake Bay, however, are conspicuously different from this general geochemical behavior normally encountered in oxygenated or moderately low oxygen environments[8, 9, 47]. DIC not only show a large enrichment against the conservative mixing line, but TA is also substantially enriched; with the excess DIC and TA reaching $275.3 \pm 59.5$ and $167.3 \pm 54.2$ µmol kg$^{-1}$ respectively (Fig. 3a, Methods).

In any estuary, the most important internal sources of TA and DIC are aerobic respiration (AR), SR, and carbonate dissolution (CD)[12, 13, 21, 45] (Fig. 4a, Table 2). Because each of these processes has a distinctly different ΔTA to ΔDIC ratio and involves a different pH change (Table 2), ΔTA:ΔDIC ratio and pH change become diagnostic of the geochemical processes. Based on the mixing line prediction, we can calculate the initial DIC and TA values at salinity $(S) = 10$ g kg$^{-1}$ for surface water and $S = 20$ g kg$^{-1}$ for bottom water. From the solubility constants we can also determine the initial concentrations of $O_2$ in $S = 10$ and $20$ g kg$^{-1}$ waters. Assuming the bottom water starts with a fully saturated dissolved $O_2$, we can then derive DIC and TA generations and pH change for each step (Table 2). In this poorly buffered water (Fig. 3b), the complete use of $O_2$ solely for AR would drive bottom-water TA and pH lower than the observed values (Fig. 4a, b). Sulfate reduction and $CaCO_3$ dissolution must then be invoked to explain the observed TA and pH. The effects of $SO_4^{2-}$ reduction on TA and DIC can be estimated from the observed [$H_2S$] (Table 2) and the rest is made up by $CaCO_3$ dissolution (Methods). We envision that these three processes can occur either sequentially (Table 2) or simultaneously when $O_2$, pH, carbonate mineral saturation state are sufficiently low. While the sequential pathway simulates the general patterns of the TA and DIC relationship (Fig. 4a) and the pH and $O_2$ relationship

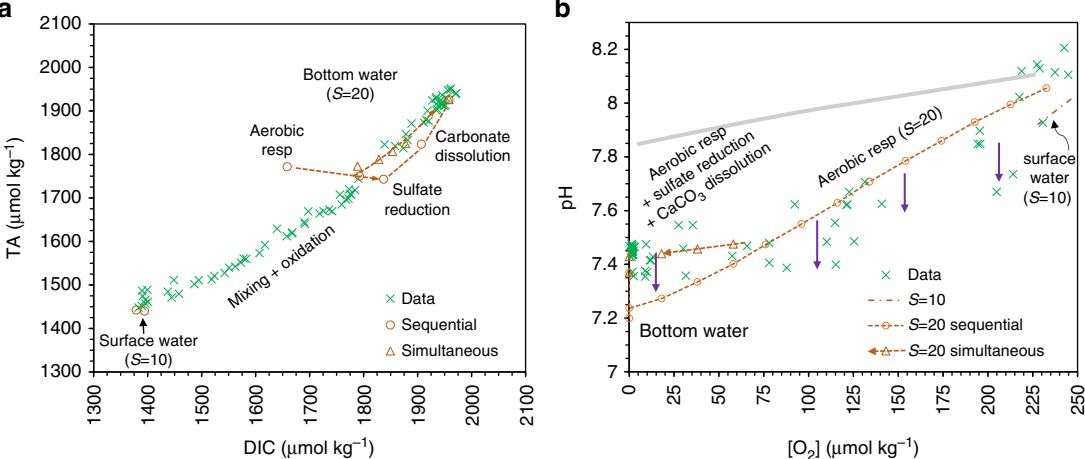

**Fig. 4** Model simulations of the evolution of total alkalinity, dissolved inorganic carbon, pH, and $O_2$ during biogeochemical processes. **a** Total alkalinity (*TA*) and dissolved inorganic carbon (*DIC*) relationship and **b** pH (25 °C and NBS scale) and [$O_2$] relationship. The biogeochemical pathway, or TA and DIC loci, includes aerobic respiration (*AR*), sulfate reduction (*SR*) and $CaCO_3$ dissolution (*CD*). Predicted pH or TA values are provided together with observed pH or TA (calculated from pH and DIC data from August 2013). Both sequential (*open circles*) and simultaneous (*triangles*) pathways are presented. *Purple arrows* represent acidification generated by oxidation of reduced chemicals ($H_2S$). The *light shaded line* at the top indicates the pH and [$O_2$] relationship from the Gulf of Mexico reflecting aerobic respiration

(Fig. 4b) reasonably well, it appears that $CaCO_3$ dissolution must have proceeded and occurred simultaneously with AR and SR as the simultaneous pathway simulates the observation better (Fig. 4a, b) and as is justified by the very low aragonite carbonate saturation state once more than 50% of $O_2$ is consumed (Table 2).

From the above simulations, we conclude that up to ~70% of the bottom-water TA production comes from $CaCO_3$ dissolution, which raises bottom-water pH from expected 7.25 to 7.45 and provides an important buffer mechanism in bottom waters (Methods). It has been reported that eutrophication led to lower pH in the polyhaline ($S > 18$ g kg$^{-1}$) part of the bay between 1985 and 2008[26]. Based on our data and model simulations, we suggest that eutrophication in the bay has led to more $O_2$ consumption, $SO_4^{2-}$ reduction, pH decrease, and dissolution of $CaCO_3$ shells and abiotic minerals in subsurface and bottom waters, consequently leading to possibly more TA and DIC export to the coastal ocean.

While shellfish calcification can represent a significant store of $CaCO_3$ in the Chesapeake Bay[26, 48], much of the CD present in the current study likely also comes from abiotic precipitation in surface waters, a mechanism noted before in the Loire estuary[49]. The extent of CD in deep waters estimated here could be supported by independently estimated $CaCO_3$ production in surface waters from deviations from conservative mixing (Supplementary Fig. 10)[48], which is consistent with the TA deficit observed here in low-salinity surface waters (Fig. 3a). While the precipitation may largely be driven by seasonal dynamics in primary production enhanced by estuarine eutrophication[49], importantly, current and future increasing atmospheric $CO_2$ due to fossil fuel production may lower surface water carbonate saturation state enough to decrease mineral formation and thus delivery below the pycnocline. If so, the bay's deep water would have a reduced capacity to neutralize metabolically generated $CO_2$, further enhancing eutrophication driven acidification.

Another important metabolic pathway is denitrification which uses $NO_3^-$ as the oxidant for organic matter decomposition[50, 51]. Note that [$NO_3^-$] is generally low in the mid-bay (<1 μmol kg$^{-1}$). However, denitrification is often coupled to nitrification at the sediment water interface. System-wide integrated denitrification rate has been estimated to be about 70 μmol m$^{-2}$ h$^{-1}$ in the

Chesapeake Bay (summer time)[52], although other estimations are lower. Taking this value as the upper end, we estimate that denitrification can contribute to a DIC production of up to 17 μmol kg$^{-1}$ and TA production of up to 16 μmol kg$^{-1}$ in a 10 m bottom-water column and over a 100-day period. This amount is only up to about 8% of the total TA production in the bottom water observed here. Finally, while organic matter decomposition using metal oxides as oxidants is important intermediate steps for biogeochemical cycles, the contributions to alkalinity production must be lower in the bay as recycled [$Mn^{2+}$] (<7 μmol kg$^{-1}$) and [$Fe^{2+}$] (<2 μmol kg$^{-1}$)[39] are much lower than the observed TA production in the bottom water; a conclusion similar to that derived in the Baltic Sea[13, 21].

## Discussion

The buffering capacity reflects the marine carbonate system's ability to resist changes in pH (or $pCO_2$) when DIC and/or TA are altered by physical and biogeochemical processes and when relevant thermodynamic constants are altered by temperature (*T*) and salinity (*S*) changes[53–58]. Mathematically, an aquatic system's ability to resist pH change can be deconstructed into its sensitivity to changes in *T*, *S*, DIC, and TA.

$$\text{dpH} = \left(\frac{\partial \text{pH}}{\partial T}\right) \text{d}T + \left(\frac{\partial \text{pH}}{\partial S}\right) \text{d}S + \left(\frac{\partial \text{pH}}{\partial \text{DIC}}\right) \text{dDIC} + \left(\frac{\partial \text{pH}}{\partial \text{TA}}\right) \text{dTA} + \ldots \tag{5}$$

Here the first and second terms represent the effects of change in thermodynamic constants as a function of *T* and *S*. The third term reflects the pH change when DIC is added while keeping *T*, *S*, and TA constant and the fourth term reflects the pH change when a strong acid ($H^+$ or $-\Delta$TA) is added while keeping *T*, *S*, and DIC constant. The slopes in the third and fourth terms are directly related to the buffer factors $\beta_{DIC}$ and $\beta_{TA}$ defined before[53, 54] with

$$\beta_{DIC} = -(2.3 \times \partial \text{pH}/\partial \text{DIC})^{-1} \tag{6}$$

**Table 2 $O_2$ consumption and DIC and TA production during sequential aerobic respiration, sulfate reduction and carbonate mineral dissolution in bottom waters[45, 83]**

| Redox | $\Delta[O_2]$ ( $\mu$mol kg$^{-1}$) | $\Delta$DIC ( $\mu$mol kg$^{-1}$) | $\Delta$TA ( $\mu$mol kg$^{-1}$) | $\Delta$TA/$\Delta$DIC | pH | $\Omega_{arag}$ |
|---|---|---|---|---|---|---|
| AR | $(CH_2O)_{106}(NH_3)_{16}(H_3PO_4) + 106O_2 \leftrightarrow 106CO_2 + 16HNO_3 + H_3PO_4 + 122H_2O$ | | | $-(16+1)/106 = -0.16$ | | |
| | 0 (100%) | 0 | 0 | | 8.055 | 1.56 |
| | −58 (75%) | +45 | −7.2 | | 7.859 | 1.05 |
| | −117(50%) | +90 | −14.4 | | 7.629 | 0.65 |
| | −174(25%) | +134 | −21.5 | | 7.421 | 0.40 |
| | −231.7(0%) | +178.0 | −28.5 | | 7.237 | 0.26 |
| SR | $(CH_2O)_{106}(NH_3)_{16}(H_3PO_4) + 53SO_4^{2-} \rightarrow 106HCO_3^- + 53H_2S + 16NH_3 + H_3PO_4$ | | | $(106+16-1)/106 = 1.142$ | | |
| | 0 | +70.0 | +80.6 | | 7.199 | 0.25 |
| CD | $CaCO_3 + CO_2 + H_2O \rightarrow Ca^{2+} + 2HCO_3^-$ | | | 2/1 | | |
| | 0 | +57.6 | +115.3 | | 7.368 | 0.38 |
| Total | −231.7 | +305.6 | +167.3 | | | |

AR, aerobic respiration; CD, carbonate dissolution; DIC, dissolved inorganic carbon; SR, sulfate reduction; TA, total alkalinity. The last columns listed the expected pH and aragonite mineral saturation state ($\Omega_{arag}$) values at the end of each step. pH and $\Omega_{arag}$ values are also calculated at the initial and mid-points of $O_2$ consumption (or % of $O_2$ saturation). For simultaneous reactions at low pH and $\Omega_{arag}$, see the text. The calculation steps and results are detailed in the Methods. Note for a more stable carbonate mineral, calcite, $\Omega_{calcite} = 1.5 \times \Omega_{arag}$

and

$$\beta_{TA} = -(2.3 \times \partial pH/\partial TA)^{-1}. \qquad (7)$$

In estuarine conditions, because $\beta_{DIC}$ and $\beta_{TA}$ are similar in magnitude[54] (also see Fig. 3b), the overall contribution to acidification or pH decrease is largely decided by changes in DIC and TA during physical and biogeochemical processes (e.g., at constant $T$ and $S$) and also by the initial buffering capacity (e.g., at variable $T$ and $S$).

The pH and $[O_2]$ relationship in Chesapeake Bay waters differs greatly from that observed in northern Gulf of Mexico (nGOM) waters (*shaded line* in Fig. 4b)[9]. It appears that Chesapeake Bay waters are more vulnerable to both anthropogenic $CO_2$ and biological induced acidifications because they have a lower buffering capacity than that of the offshore waters, in particular, in the nGOM as TA and DIC are lower in the Susquehanna River and US eastern margin waters than those of the Mississippi River and nGOM seawater (Figs. 3b, 4b and 5a)[9, 59, 60]. However, our simulations and those of the previous studies[12, 53–57] suggest that lower buffering capacity itself does not necessarily lead to low pH (Fig. 5); rather, it allows a much greater pH decrease when other sources of $CO_2$ or strong acids are added (Fig. 5b). Similar amounts of AR (Table 2) would lead to a pH decrease of only 0.4 units in the strongly buffered nGOM waters whereas a larger decrease of nearly 0.8 units would occur in the poorly buffered Chesapeake Bay waters at the present day conditions (at $S = 34$ and 20 g kg$^{-1}$, respectively, Figs. 4b and 5c).

While the OA signal due to $CO_2$ uptake in the open ocean regions is similar across middle and lower latitudes, the manifestation of this anthropogenic $CO_2$ signal through ocean–river mixing in estuaries is dependent on the river TA and DIC values, which are highly variable among the world's rivers[61], and whether $CO_2$ is also introduced via microbial respiration[11, 60]. Due to the very high river TA and DIC and the resulting strong buffering capacity over the entire salinity range in the Mississippi River impacted coastal waters, pH change due to OA is proportional to the open ocean OA source signal and salinity and decreases toward zero salinity (Fig. 5a)[9, 60, 62]. In the Chesapeake Bay where average river TA, DIC, and buffering capacity are low, however, the oceanic OA signal is amplified in the low and middle salinity zone. Here the combination of reduced buffering capacity (with decreasing salinity) and a still sufficiently strong open ocean OA signal generates a minimum buffer zone[60] and thus a Maximum Estuarine Acidification Zone (MEAZ) (Fig. 5c). The existence of a MEAZ and its salinity range depend not only on the river TA

value, but also the TA:DIC ratio[60]. When $CO_2$ addition from AR increases from 0 to 100 and finally to ~200 $\mu$mol kg$^{-1}$ (or 0 to roughly half or to a full $O_2$ consumption depending on the salinity and temperature), the minimum buffer zone shifted from salinity ~4 to ~13 (Fig. 5a) and finally to ~23 (Fig. 5b)[60]. Note that local $CO_2$ uptake, carbonate mineral dissolution and SR are not included in this discussion (Fig. 5) and would further modify the estuarine buffering capacity as they would modify the TA to DIC ratio in estuarine waters (Table 2, Figs. 3b, 4a, b).

Below we further discuss the effects of anthropogenic $CO_2$ and biological $CO_2$ and acid additions on estuarine pH buffering capacity[53, 54]. The marine carbonate system has a minimum buffering or maximum pH change point when DIC increases approximately equal to that of TA (or TA:DIC $\approx$ 1) where $[CO_2]$ $= [CO_3^{2-}] + [B(OH)_4^-]$ (if we ignore all other weak acid–base species). At this point, any addition or removal of $CO_2$ or acids will result in a maximum pH decrease or increase. Because DIC is slightly higher than or nearly equal to TA in rivers[47, 62, 63] and is lower than TA in seawater, there may exist a small crossover of DIC and TA at the very low-salinity zone. A peculiar pH minimum occurring in the low-salinity zone of estuaries is related to this mixing feature as was noticed a long time ago in both closed[64] and open[65] system simulations. Furthermore, how $CO_2$ is added to the estuarine waters affect how the crossover point will move. First we contend anthropogenic $CO_2$ does not directly add to the high $pCO_2$ river water but is mixed into the estuary via river–ocean mixing[60]. In contrast, respiratory $CO_2$ is nearly equally added to the bottom water based on $O_2$ consumption regardless of the mixing index or salinity (except that $O_2$ solubility increases when salinity decreases, but it is a small correction). In Fig. 6, we summarize several scenarios illustrating how the crossover point of the TA and DIC to salinity lines or the point of TA:DIC ratio = 1 moves along the TA-salinity line. Adding anthropogenic $CO_2$ to the seawater endmember would move this crossover point to only a slightly higher salinity. Adding biological $CO_2$ (for example 100 $\mu$mol kg$^{-1}$) to both the river and ocean endmembers would, however, shift the DIC line to a much higher position (parallel to the original line) creating a crossover point located at a salinity substantially higher than the original one. Finally the combined effect of anthropogenic $CO_2$ and biological $CO_2$ from respiration moves the crossover point to an even higher salinity. These crossover points are consistent with the progressive shift of the minimum buffer factor ($\beta_{DIC}$), the pH minimum, and the maximum acidification zone ($-\Delta$pH) presented in Fig. 5. However there appears a difference in the location (salinity) between the minimum buffer factor ($\beta_{DIC}$) and

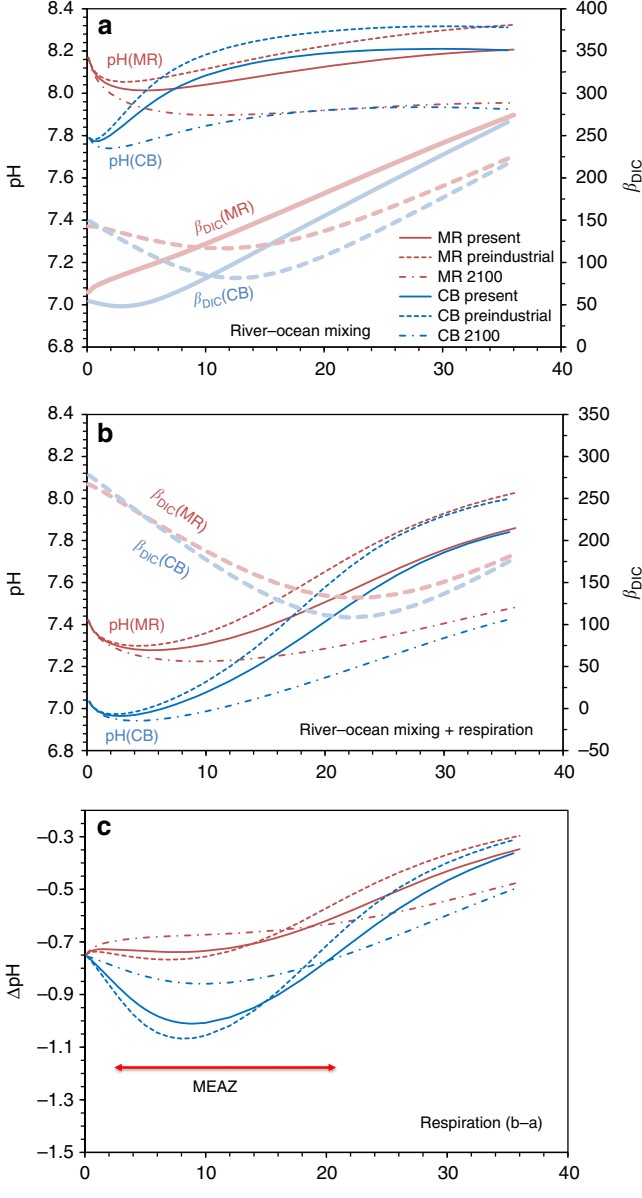

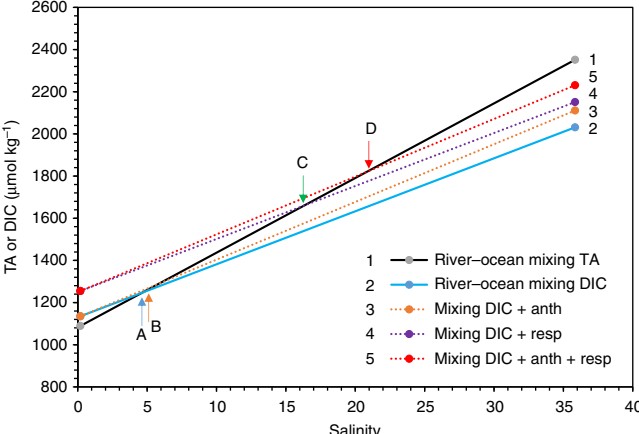

**Fig. 6** Estuarine TA or DIC mixing lines between the river and ocean endmembers with additional $CO_2$ added in from anthropogenic sources and respiration. Line 1 is river-ocean mixing of TA, line 2 is river-ocean mixing of DIC, line 3 is mixing DIC plus anthropogenic $CO_2$ addition, line 4 is mixing DIC plus respiratory $CO_2$ addition, and line 5 is mixing DIC plus both anthropogenic and biological $CO_2$ additions. The crossover points are river-ocean mixing lines of TA with A river-ocean mixing lines of DIC, B mixing plus anthropogenic $CO_2$, C mixing plus $CO_2$ from respiration, and D mixing plus $CO_2$ from both B and C

**Fig. 5** Simulations of pH and pH decrease and the evolution of the buffer factor $\beta_{DIC}$ in Chesapeake Bay (*CB*) waters and Mississippi River (*MR*) coastal waters. **a** pH curves (*thin lines*) during estuarine mixing under three scenarios (preindustrial era, present day, and year 2100). Buffer factor $\beta_{DIC}$ for the present day condition (*thick solid lines*) and the present day condition plus 100 μmol kg$^{-1}$ (*thick dashed lines*) are also presented for both estuaries. **b** pH curves and $\beta_{DIC}$ curves plus full $O_2$ consumption (respiration) during estuarine mixing. (**c**) pH decrease (−ΔpH) as a result of full $O_2$ consumption during estuarine mixing (a difference between **b** and **a**). The *red arrows* represents the Maximum Estuarine Acidification Zone (*MEAZ*). Note that $\beta_{DIC}$ presented here does not match that from the real data (Fig. 3b) at low salinities as a stratified and $O_2$ depleted state does not exist for the very low salinity part of the Chesapeake Bay

the maximum acidification (−ΔpH). This is because ΔpH represents the accumulative pH change between an end point and a beginning point while the buffer factor reflects the pH sensitivity at a specific point if additional DIC is added to the system.

In summary, large estuarine water bodies, exemplified by the Chesapeake Bay, are particularly vulnerable to the anthropogenic $CO_2$ and eutrophication-induced coastal OA. In this paper we

emphasize that subsurface $H_2S$ oxidation (~0.2 pH units) and local surface $CO_2$ uptake (~0.1 pH units) work together with known organic matter respiration and the open-ocean OA source signal to drive substantial acidification and $CaCO_3$ dissolution in estuarine subsurface waters. Currently, acidification due to $CO_2$ input from AR (up to 0.8 pH units) in the bay exceeds that from the atmospheric $CO_2$ increase in the open ocean (~0.1 pH units in surface waters and the signal is mixed into estuaries proportional to salinity) and local estuarine uptake; but towards the end of this century the latter will approach or exceed the former and the synergy between them will also increase. In addition, future increasing atmospheric $CO_2$ due to fossil fuel production may lower carbonate saturation state enough to decrease mineral formation in surface water and thus delivery below the pycnocline, where we have currently demonstrated that $CaCO_3$ dissolution offsets a significant proportion of the metabolic $CO_2$ effect on acidification. We further recognize that natural and anthropogenic acidification mechanisms most relevant to estuarine acidification are characterized by various time scales. They range from nearly instantaneous for acid–base equilibrium[66], to minutes for $CO_2$ hydration[66], and to minutes to hours for $H_2S$ oxidation[36, 41, 67]. In contrast, time scales for physical mixing are on the order of tidal or less, to daily and seasonal[30, 31, 68] while local $CO_2$ uptake from the atmosphere and its accumulation in the water column and acidification of the bottom water as well as pelagic and benthic respirations operate over tidal to seasonal scales[12, 13, 35, 37, 69]. Although anthropogenic changes in external forcing due to variability in river and ocean endmembers may also have a seasonal component, decadal and centennial variation is more important[5, 35, 70, 71]. This mosaic of processes with different time and space scales poses a great challenge in our ability to understand and predict coastal OA.

## Methods

**Site and cruise descriptions**. The Chesapeake Bay is the largest estuary in the US. The August 2013 survey started from the upper estuary near the Susquehanna River mouth (Fig. 1a). The upper and middle Chesapeake Bay were surveyed during 9–14 August 2013 by RV *Hugh R. Sharp*. The water column survey focused mainly at one site south of the Bay Bridge (38°58.8 N, 76°22 W), where a field study

of redox chemistry[39] and water column inorganic carbon and pH were carried out. We consider this site as the up end of the mid-bay. We repeatedly sampled the water column at high slack tide and low slack tide. During this 5-day survey, an excursion was made south to the middle bay near Solomons Island. After the completion of the cruise, we set up two 24-h dockside measurements, one at the dock of the Chesapeake Biological Laboratory (CBL; 38.317317° N, 76.450980° W) on Solomons Island near the southern end of our ship-based survey and another further south at a pier of the Virginia Institute of Marine Science (VIMS; 37.2473° N, 76.4994° W) near the bay mouth. A similar study was conducted during 18–24 August 2014 (Supplementary Fig. 1). We also conducted a spring survey (11–16 April 2015) to get an initial condition before the hypoxia season (Fig. 1b).

**Sample and analytical methods.** Salinity, temperature, and $O_2$ were obtained from the CTD Rosette system. Total sulfide ($H_2S$ and $HS^-$) was determined by voltammetry using solid state Au/Hg electrodes[36, 39, 43]. Surface water partial pressure of $CO_2$ ($pCO_2$), position, temperature, and salinity information were measured underway while the ship was sailing or anchored by pumping surface water from under the ship to the shipboard laboratory using an underway $pCO_2$ system[72]. TA and DIC water samples were taken from Niskin bottles and were preserved and stored in 250 ml borosilicate glass bottles with 100 μl saturated $HgCl_2$ solution[73]. TA and DIC samples were stored at refrigerated temperature (~5 °C) before being measured (within 4 weeks). TA samples were measured by open-cell Gran titration with a precision better than ±0.1% using an Apollo Scitech Seawater Total Alkalinity titration system[73]. DIC samples were analyzed by adding phosphoric acid into sample waters to release $CO_2$, which was measured by an infrared $CO_2$ analyzer (LI-COR 7000) with an overall precision of ±0.1% using an Apollo Scitech DIC Analyzer[73]. Both TA and DIC measurements were quality controlled by Certified Reference Materials from Andrew Dickson of the University of California at San Diego. pH samples were taken by the same Niskin bottle and were measured by an Orion Ross glass electrode within 1 h after the water temperature was stable in a 25.0 ± 0.1 °C thermal bath on the research vessel. The electrode was calibrated against three NBS (NIST) standards and pH values are reported in NBS scale and at 25 °C. Note that pH values in NBS scale are about 0.1 pH unit higher than those reported in total $H^+$ scale ($pH_T$) elsewhere.

**Uncertainty in determining DIC and TA enrichment.** We averaged all subsurface and bottom waters ($S > 11$ g kg$^{-1}$) to derive a DIC enrichment of 275.3 ± 59.5 and a TA enrichment of 127.3 ± 54.2 μmol kg$^{-1}$, with respect to their expected conservative behaviors. The TA enrichment is underestimated because of a technical challenge caused by HgS precipitation, which releases $H^+$ when $HgCl_2$ was used to stop microbial activity[18], and/or oxidation of $H_2S$ and $NH_4^+$ during sample storage and/or analysis (which also generate $H^+$). TA values calculated from DIC and pH analyzed onboard (neither is subject to the sample preservation and storage problems) agree well with measured TA except in the bottom waters (Supplementary Fig. 11a), and the disagreement increases as [$H_2S$] increases (Supplementary Fig. 11b). The internal consistency analysis suggests that TA reduction due to sample preservation and storage is 40 ± 20 μmol kg$^{-1}$; thus, the most likely TA enrichment in the bottom water is (127.3 ± 54.2) + (40 ± 20) = 167.3 ± 57.6 μmol kg$^{-1}$.

**Determination of the endmembers and mixing lines.** Lowest salinity values (from the station immediately downstream of the Susquehanna River) were selected as the river endmember values ($S$, TA, and DIC were measured at 0.189 g kg$^{-1}$, 1089.2 μmol kg$^{-1}$, and 1115.1 μmol kg$^{-1}$, respectively), though the station near the canal had the lowest TA and DIC due to mixing of water from the Delaware Bay. The offshore end-member data were collected at Latitude and Longitude of 37.13333° N and 73.32533° W on August 14 at the end of the cruise. We took the average values of the surface 45 m as the ocean endmember ($S = 33.618 ± 0.139$ g kg$^{-1}$, TA = 2248.4 ± 78.4 μmol kg$^{-1}$, and DIC = 2037.9 ± 72.4 μmol kg$^{-1}$). So, the TA conservative mixing line is TA = 34.676 × $S$ + 1082.7 and the DIC line is DIC = 27.607 × $S$ + 1109.9.

**Calculation of air–sea $CO_2$ flux and the DIC increase.** Water surface $pCO_2$ was measured every 1–1.5 min with calibrations every 6–12 h (August 2013, August 2014, and April 2015). Atmospheric $pCO_2$ values were also measured every 2–4 h during these cruises. Both atmosphere and water $CO_2$ values were measured in a dry condition ($xCO_2$) and were converted to $pCO_2$ in 100% water saturated conditions inside the equilibrator ($pCO_{2(eq)}$) by considering water vapor pressure:

$$pCO_{2(eq)} = xCO_{2(eq)} \times (P_b - P_{weq}) \tag{8}$$

where $P_b$ is barometric pressure and $P_{weq}$ is water vapor pressure in the equilibrator. For water data, the $pCO_{2(eq)}$ is further converted to estuarine surface water $pCO_2$ ($pCO_{2(water)}$) by considering temperature changes between the surface water and the equilibrator through the following equation[72]:

$$pCO_{2(water)} = pCO_{2(eq)} \times \exp(0.043 \times (SST - T_{eq})) \tag{9}$$

where SST is sea surface temperature (°C) and $T_{eq}$ is temperature in the equilibrator. Our measured atmospheric $xCO_2$ values were also converted from dry

condition to near sea surface wet condition ($pCO_{2(air)}$) by Eq. (10):

$$pCO_{2(air)} = xCO_{2(eq)} \times (P_b - P_w) \tag{10}$$

Here $P_b$ is barometric pressure and $P_w$ is water vapor pressure at the sea surface. Each $pCO_{2(water)}$ and its corresponding $pCO_{2(air)}$ were used to calculate the gas exchange flux ($F_{CO_2}$) between atmosphere and water by Eq. (11).

$$F_{CO2} = C_2 \times k \times K_0 \times (pCO_{2(water)} - pCO_{2(air)}) \tag{11}$$

where $k$ represents the gas transfer velocity and $K_0$ is the solubility of $CO_2$[74]. We adopted Ho et al.[75] as the gas transfer velocity and an ensemble of gas transfer parameters to evaluate the uncertainty range following the previous practice[72, 76]. Finally, the coefficient $C_2$ corrects the non-symmetrical distribution of wind[76]. A negative air–sea $CO_2$ flux means an uptake of atmospheric $CO_2$ for the water.

Over at least a 100-day water residence period (from May to August) and a water column of 20 m, this $CO_2$ flux can be converted into an increase in DIC of, 4.3–6.7 (mmol m$^{-2}$ d$^{-1}$) × 100$d$/20 m ≈ 21.5–33.5 mmol m$^{-3}$ or 21.2–33.1 μmol kg$^{-1}$ (here a density of 1012.09 kg m$^{-3}$ is used). We used the entire water column rather than the surface mixed layer because the main concern here is how local $CO_2$ uptake, via internal mixing, contributes to acidification of the especially vulnerable bottom waters.

**Calculation of pH decrease due to local $CO_2$ uptake.** With $H_2S$ included, the calculation of pH (in NBS scale and at 25 °C) decrease was performed using the modified CO2SYS program. Note another program, AquaEnv, also has such a capacity[77]. We used day 4 data with $S = 18.618$ g kg$^{-1}$, $T = 25.28$ °C, depth = 17 m, DIC = 1933.8 μmol kg$^{-1}$, total [$H_2S$] = 37.79 μmol kg$^{-1}$, [$NH_3$+$NH_4^+$] = 13.6 μmol kg$^{-1}$, T-$PO_4$ = 3.5 μmol kg$^{-1}$, and pH = 7.476 to calculate a TA = 1934.1 μmol kg$^{-1}$. Then, we subtracted the summer DIC by 21.2–33.1 μmol kg$^{-1}$ (=1912.6–1900.7 μmol kg$^{-1}$) to calculate a new pH (7.557–7.604). Thus, the pH decrease by an increase of DIC derived from local uptake of atmospheric $CO_2$ is 0.081–0.128 pH unit over the entire period from spring to summer.

**pH decrease due to oxidation of reduced chemicals.** We used day 4 data with $S = 15.145$ g kg$^{-1}$, $T = 25.36$ °C, depth = 12.54 m, DIC = 1767.7 μmol kg$^{-1}$, [$H_2S$] = 2 μmol kg$^{-1}$, and pH = 7.354 to calculate a TA = 1698.8 μmol kg$^{-1}$. Then, we subtracted a 30–60 μmol kg$^{-1}$ from TA to calculate a new pH (7.152–7.246). Thus, the pH decrease by a 30–60 μmol kg$^{-1}$ of TA reduction is 0.108–0.202 pH units (represented by the *purple arrows* in Fig. 4b). The modified version of CO2SYS was used for all the $CO_2$ and pH calculations. Note, adding a <3 μmol kg$^{-1}$ of T-$PO_4$ would only lead to <0.005 pH unit decrease in the calculation. Thus, its influence is ignored here.

**Modification of the CO2SYS program.** The modifications were done on the Excel version 2.1 of the program[78], which is available for download from CDIAC (http://cdiac.ornl.gov/ftp/co2sys/). A Matlab version is available from the corresponding author. In addition to the total Phosphate and Silicate, the program now accepts the total $NH_3$ and total $H_2S$ in μmol (kg of SW)$^{-1}$. The contribution of each to the alkalinity is given by:

$$NH_3 - Alk = [NH_3]_T \frac{K_{NH_3}}{K_{NH_3} + [H]} \tag{12}$$

and

$$H_2S - Alk = [H_2S]_T \frac{K_{H2S}}{K_{H2S} + [H]} \tag{13}$$

where $K_{NH_3}$ and $K_{H2S}$ are the dissociation constants of ammonium ($NH_4^+$) and hydrogen sulfide ($H_2S$).

The dissociation constant for $NH_4^+$ was taken from Clegg and Whitfield[79] and is valid for $S = 0$–40 g kg$^{-1}$ and $t = -2$ to 40 °C (note ref. [21] essentially provided the same constant). The constant for $H_2S$ was taken from Millero et al.[80] and is valid for $S = 0$–40 g kg$^{-1}$ and $t = 0$–35 °C. When the pressure is not zero, a correction is applied according to Millero[81]. A comparison with AquaEnv under [$H_2S$] <50 (or 300) μmol kg$^{-1}$ shows a good agreement of calculated pH (in free scale) within 0.0003 (or 0.0026) from known TA and DIC. We have further tested the calculations with waters containing high concentrations of $H_2S$ and posted this modified version of the CO2SYS program on the CDIAC website for public access[82].

**Simulation of bottom-water geochemical pathways.** We present here the calculation methods for Table 2 and Fig. 4a, b. For the bottom-water condition ($S = 19.87$ g kg$^{-1}$ and $T = 25$ °C), we have a saturated [$O_2$] = 231.7 μmol kg$^{-1}$, and, from the mixing line at $S = 19.87$ g kg$^{-1}$, TA = 1771.1 μmol kg$^{-1}$ and DIC = 1658.5 μmol kg$^{-1}$, we have pH = 8.066. If all dissolved $O_2$ is used by heterotrophic bacteria for organic carbon respiration via Redfield stoichiometry[45, 83], it would increase DIC by 178.0 μmol kg$^{-1}$ and decrease TA by 28.5 μmol kg$^{-1}$. To be consistent with our observation, SR and $CaCO_3$ dissolution must have increased bottom-water TA by a

total of 195.9 µmol kg$^{-1}$ (i.e., observed 167.3 µmol kg$^{-1}$ plus expected −28.5 µmol kg$^{-1}$) beyond conservative mixing. We estimate TA increase from $SO_4^{2-}$ reduction as 80.6 µmol kg$^{-1}$ from the total concentration of $H_2S$ (35 µmol kg$^{-1}$)[39] by the following equation:

$$\Delta TA = (2 \times [H_2S] + 16/53 \times [H_2S]) \qquad (14)$$

where $2 \times [H_2S]$ represents an equal amount of $HCO_3^-$ and $HS^-$ production during $SO_4^{2-}$ reduction and $16/53 \times [H_2S]$ represents $NH_3$ production (and contribution to TA) based on stoichiometry (see Table 2). Then the TA generated from $CaCO_3$ dissolution must be as high as 115.3 ± 20.0 µmol kg$^{-1}$ by the difference (195.9–80.6) and contributes up to 70% of total amount of TA production. The amount of DIC production following these steps is 305.6 ± 10.0 µmol kg$^{-1}$. This is within the uncertainty of the observed value of 275.3 ± 59.5 (Fig. 3). The 10% difference (30 µmol kg$^{-1}$) can be explained either by TA increase due to organic matter respiration using nitrate (denitrification) and metal oxides[13, 45] and/or the deviation of C/N ratio from the Redfield ratio[13, 84, 85] as well as probably organic alkalinity contribution[86]. Indeed if a lower C to $-O_2$ ratio (106/154 = 0.688) given in ref. [84] is used, the produced DIC would be close to the observation.

From the resulting pH and $\Omega_{arag}$ (Table 2), it is clear when $O_2$ is partially consumed aragonite mineral becomes undersaturated (starting at 75% $O_2$ saturation) and $CaCO_3$ dissolution can proceed together with AR. To simulate the observed DIC (measured) and TA (calculated from DIC and pH) data, we assume the dissolution does not occur until a sufficiently low $\Omega_{arag}$ in waters and that the first 20% of $CaCO_3$ dissolution occurs before or at DO = 33%. Then the second, third and fourth 20% of the $CaCO_3$ dissolution occurs before or at DO = 16%, 8%, and 0%, respectively. The last 20% of the $CaCO_3$ dissolution occurs together with SR.

**Simulation of pH changes**. We present here the calculation methods for Fig. 5a, b. Although each term in Eq. (5) may be derived analytically, in this paper, we obtain the overall pH change, ΔpH (presented in Fig. 5c), numerically using the updated CO2SYS program. Conditions used are given below.

We assume a present day atmospheric dry $CO_2$ fraction ($xCO_2$) as of 396.2 ppm and the corresponding water $pCO_2$ = 384.0 µatm at 25 °C and salinity = 36.0 g kg$^{-1}$. $xCO_2$ is set to 281.3 ppm for the pre-industrial era and 798.0 ppm for year 2100.

For the Chesapeake Bay simulation, we take TA from the offshore water at 74.9 m, $S$ = 35.839 g kg$^{-1}$, and TA = 2351.5 µmol kg$^{-1}$. Equilibration of this water with the atmosphere yield a DIC = 2030.3 µmol kg$^{-1}$ and pH = 8.198 (in NBS scale) at 25 °C and 1 m of water depth. For the Gulf of Mexico water, we take the endmember values from Cai et al.[9] and adjust the present condition slightly to the above $pCO_2$. The present day conditions are $S$ = 36.3, TA = 2398.1 µmol kg$^{-1}$, DIC = 2065.2 µmol kg$^{-1}$, pH = 8.203 at 25 °C and 1 m water depth.

For the Mississippi River end-member conditions, we have $S$ = 0.1 g kg$^{-1}$, TA = 2400 µmol kg$^{-1}$, DIC = 2430 µmol kg$^{-1}$, and $pCO_2$ = 1388.9 µatm. For the Susquehanna River, we have $S$ = 0.189 g kg$^{-1}$, TA = 1089.2 µmol kg$^{-1}$, DIC = 1135 µmol kg$^{-1}$, and $pCO_2$ = 1514.9 µatm.

For Mississippi/GOM bottom water, at $S$ = 34 g kg$^{-1}$, $T$ = 25 °C, and $[O_2]$ = 216.0 µmol kg$^{-1}$, complete $O_2$ consumption would lead to a DIC increase of 216 × 106/138 = 165.9 µmol kg$^{-1}$ and a TA decrease of 26.6 µmol kg$^{-1}$. For Chesapeake Bay bottom water, at $S$ = 20 g kg$^{-1}$, $T$ = 25 °C, and $[O_2]$ = 232.6 µmol kg$^{-1}$, complete $O_2$ consumption would lead to a DIC increase of 216 × 106/138 = 178.7 µmol kg$^{-1}$ and a TA decrease of 28.7 µmol kg$^{-1}$.

**Buffer factor calculation**. We calculate these buffer factors (Fig. 5) following the analytical formula provided by Egleston et al.[54] with a typo corrected[87]. Specifically, we extract out species concentrations and thermodynamic constants from the updated version of CO2SYS. We also compare results with those calculated with two other programs AquaEnv and SEACARB (http://CRAN.R-project.org/package=seacarb) and the agreement is reasonable (within 8 µmol kg$^{-1}$ or 3%) as is expected. Although $H_2S$ and $NH_4$ are not included in the analytical equations, as our equilibrium calculation already include these acid–base species and $[H_2S]$ and $[NH_4]$ are not high in the bay, the calculated results are similar to those from the AquaEnv which includes fully these reduced chemical species. Finally, in Fig. 3b, for the real system buffer factors, we directly use AquaEnv. Given the low $[H_2S]$ in the bay, with or without $H_2S$ have only resulted in a minor difference in buffer factor calculation.

**Computer program availability**. The modified CO2SYS program on Excel version 2.1 is available for download from CDIAC (http://cdiac.ornl.gov/ftp/co2sys/). The Matlab version is available from the corresponding author upon reasonable request.

**Data availability**. All data are available from the corresponding author upon reasonable request and will be deposited at the US National Centers for Environmental Information (https://www.nodc.noaa.gov/oceanacidification/).

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

## Acknowledgements

This work was supported by internal funds from the University of Delaware Provost's office and the College of Earth, Ocean and Environment Dean's office to W.-J.C., by grants from the National Science Foundation (NSF OCE-1559312) and NASA (NNX14AM37G) to W.-J.C., by grants from NSF (OCE-1155385) and the U.S. National Oceanic and Atmospheric Administration (NOAA) Sea Grant program (NA14OAR4170087) to G.W.L., and a NOAA grant to J.T., W.-J.C., M.L., G.G.W., J.C.,

and W.M.K. (NA15NOS4780190, publication # 17-001). We thank Xinping Hu for discussion and the Chesapeake Bay Program and the Maryland Department of Natural Resources for the monitoring data. This is UMCES publication number 5369.

## Author contributions

W.-J.C. and G.W.L. are responsible for the design of the fieldwork. W.-J.C. is responsible for data analysis and writing of the paper. W.-J.H., M.X., A.J., R.M., J.B., N.H. are responsible for data collection and sample analysis. Y.-Y.X. contributes to buffer factor calculation. M.L. contributes to physical mixing part. J.T. and W.M.K. contribute to the biological production part. All authors have contributed to discussion and revision of the paper.

## Additional information

**Competing interests:** The authors declare no competing financial interests.

