## [Peer Review File · Nature Communications]

Reviewers' Comments:

Reviewer #1:

Remarks to the Author:

Thank you again to the authors for addressing and clarifying the issues addressed by myself and the other reviewers in an earlier version of this manuscript. Regarding the authors' responses, I fully agree with their response to Reviewer #2's comment on the treatment of TA generation through sulphate reduction. It is well known that whichever acid-base species of DIC, $\Sigma\text{H}_2\text{S}$, ΣPO_4 or ΣNH_4 is used to describe this (or any) reaction does not impact the generated TA. In their response, the authors have thoroughly and correctly shown that the $\Delta\text{TA}/\Delta\text{DIC}$ ratio is 1.14 irrespective of the chosen formulation for the acid-base speciation. In fact, the chosen formulation for the acid-base speciation should also not impact the net generated number of protons, as is nicely shown by Hofmann et al. (Mar Chem, 2010). I also think that the authors have generally responded well to the other comments raised by myself and Reviewer #3. Moreover, I believe that the additional space allowed by Nature Communications improves both the flow and clarity of the manuscript, and that therefore this journal is a more suitable platform for this manuscript. Given the relatively few changes between the current and previous versions, and my overall positive feeling on the current version of this manuscript, I only have some minor comments, which are given below:

Abstract: Here I have a preference for certain formulations used in the previous version of the manuscript, e.g. on line 22 I would replace "lead" with "may lead" and in line 24 I would replace "It is not clear" with "It is less clear".

Lines 42-44: I would suggest using either "...contribute to far greater acidification..." or "cause a far greater acidification effect" (or equivalent) here.

Line 106: shouldn't this be August 9, 2013?

Lines 135-137: I'm not sure if this was added because of an earlier reviewer comment, but for clarity I would suggest choosing one formulation for Eq. (1) only (possibly the one using H_2S) and removing this sentence.

Line 156: It may be better to stick to $-\Delta\text{TA}$ only here. In terms of stoichiometry, how much H^+ is generated exactly through these processes depends on the ambient pH, i.e. the relative fractions of NH_4^+ versus NH_3 and H_2S versus HS^- (see e.g. again the Hofmann et al. Mar Chem 2010 work).

Section "Vulnerability of estuarine waters to acidification": I miss a sentence in this discussion that I considered rather clarifying in the previous manuscript version: "However, our simulations and those of the previous studies suggest that lower buffering capacity itself does not necessarily lead to low pH (Fig. 5a); rather, it allows a much greater pH decrease when other sources of CO_2 or strong acids are added (Fig. 5b)." Any particular reason why this has been removed?

Lines 405-413: Am I understanding correctly that a range of DIC increases is taken here from both 2013 and 2014 data, whereas the initial conditions of the calculation are from the 2013 data only? Why aren't the pH changes for 2013 and 2014 calculated entirely independently, i.e. by using the measured initial conditions for 2014 and then subtract the 2014 DIC increase? And how is the pH change for April 2015 calculated? Shouldn't, in this case, the DIC change be added to the measured 2015 spring data?

Lines 504-505: "Given the low H_2S in the bay, in- or excluding H_2S resulted in a minor difference only"

Fig. 3b: shouldn't the same symbols be used here as in Fig. 3a? (i.e. a distinction between the

various data sources). Also, what do the bars in the upper right corner represent, and how have they been defined? If they are "end-members" like in Fig 3a, why are they bars instead of dots?

Reviewer #2:

Remarks to the Author:

In this revised manuscript, the authors investigate how interplay of natural and anthropogenic CO₂ sources, redox reactions, and sediment-water exchanges govern ocean acidification rates in the Chesapeake Bay where eutrophication, hypoxia and anoxia, and low pH occur due to anthropogenic nutrient inputs. The authors carry out in-depth analyses of good quality measurement data for oxygen (O₂), hydrogen sulfide (H₂S), pH, dissolved inorganic carbon (DIC) and total alkalinity (TA). They focus on a pH minimum observed in intermediate depths and argue oxidation of reduced chemicals in bottom water, carried upwards by mixing, is responsible for it. They provide supporting evidence, in the form of subsurface enrichment of TA and DIC for which they invoke sulfate reduction and calcium carbonate dissolution to explain. In order to further substantiate this, they show that simple numerical model incorporating the above processes together with aerobic respiration of organic matter is able to simulate the observed DIC and TA relationship in subsurface water. In conclusion, the authors suggest that synergistic effect from river-ocean mixing, global and local CO₂ uptake from the atmosphere, CO₂ and acid production from respiration, and other redox reactions lead to a poor acid buffering capacity, severe acidification and carbonate mineral dissolution in the Chesapeake Bay and likely other estuaries worldwide.

This is well written manuscript that advances our understanding of the complex processes governing OA rates in coastal ocean. To my knowledge, it is the first of its kind and represents an important contribution to coastal biogeochemistry. The authors have also addressed the earlier reviewer comments adequately. Therefore, I recommend publication in Nature Communications after minor revision in which the authors address my following specific comments.

Page 4, line 92: "are not insignificant" please change to "are significant"

Page 6, line 126-127: "...and bottom-water pH and TA were lower in 2014 than 2013 (Fig. 2a-e)" To me it seems there is a pH decrease that cannot be explained by DIC and TA. From Figs 2b and 2c it seems that bottom water DIC and TA are lower in 2014 by the same magnitude as compared to 2013 (the x-axis scale is very coarse and I might be wrong here). If so, and if S difference is not significant, why then bottom water pH_{25C} has decreased so much in 2014?

Page 15, line 334: can the authors provide any information regarding if they controlled the accuracy of the O₂ sensor?

Pages 15, 340-341: how long were the TA and DIC samples stored before analyses?

Page 16, lines 355 – 356: The use of HgCl₂ for DIC/TA sample preservation is quite normal. It is therefore of general interest that the authors report a TA decrease "caused by HgS precipitation, which releases H⁺ when HgCl₂ was used to stop microbial activity..." Is this a general "warning" about TA decrease whenever HgCl₂ is used for preservation? Or do they believe this was specific to their situation?

Finally, I notice that equations are numbered as 1,2,..both in the main text and in the Methods. For unique identification of equations, it may be wise to number them differently. For instance, M1, M2, ..in Methods.

Reviewer #3:

Remarks to the Author:

Recommendation: Accept with minor revisions

I liked the paper and I recommend publication. I have a few general comments the authors should consider along with a few minor clarification points.

The paper presents a novel study that looks at the role of eutrophication and redox reactions on the acidification of Chesapeake Bay. The observations show the existence of a mid-depth pH minimum associated with the oxidation of reduced chemicals - which is both novel and interesting. Further, they show in the anoxic bottom water there is rapid increases in TA and DIC related to SR and CD. These two observed features are linked because the SR reduction produces H₂S which is mixed up into the mid water and contributes to the observed pH minimum. I note while the buffer capacity of the Chesapeake Bay water appears low, the CD in the highly acidic bottom water introduces another way the bay can buffer future rising CO₂ levels and this is an important feature to consider in future OA projections for this Bay. I would like the authors to more clearly acknowledge this mechanism in the revising their conclusion about the future OA projections for the Bay.

My other suggestion is to redraw figure 4 to remove the salinity mixing trend to produce a TA-DIC figure that is only influenced by BGC processes. I note that SR and CD appear to have a similar slope in figure 4, is this correct because it appears graphically in the figure that it would be difficult to separate these two processes. The amount of CD seems large does this reflect a large amount of CaCO₃ in the sediments or the dissolution of recently produced CaCO₃ from the upper ocean? If it is the former process, how long before one runs out of CaCO₃ to buffer the OA impact (is this relevant to the OA projections for the Bay?)

I'm satisfied with their calculation of TA and DIC changes associated with SR, which was raised in the previous review.

Minor comments

I75, in the mid-bay state how much below atmospheric CO₂ the surface water is

I116, 6m depth still within the upper mixed layer is this correct? seems inconsistent with a large drop in pH

I212, change CB to CD

I273, the poor buffering capacity of Chesapeake Bay water is interesting but in the bottom water of the bay it is the dissolution of CaCO₃ that partially buffers the increase in CO₂ (external source for buffering). Does the low buffering capacity of the water contribute to the low CaCO₃ saturation state and do this set up the ideal conditions for high CaCO₃ dissolution?

I297, and in the conclusion, Yes! as you already showed these processes would have a significant impact on TA and DIC values. Would this substantially alter the sensitivity of the bay to future OA? it appears the Bay has a natural way to buffer OA by dissolving CaCO₃, you should discuss this if you want to comment on future OA response of the bay.

Figure 4, not clear if both the sequential and simultaneous lines start at the same initial point? Do they? if not why not?

Figure 5. state that panel c lines come from the difference between the lines shown in panels a and b.

Reviewers' comments (IN BLACK) and author responses (IN BLUE):

Reviewer #1 (Remarks to the Author):

Thank you again to the authors for addressing and clarifying the issues addressed by myself and the other reviewers in an earlier version of this manuscript. Regarding the authors' responses, I fully agree with their response to Reviewer #2's comment on the treatment of TA generation through sulphate reduction. It is well known that whichever acid-base species of DIC, $\Sigma\text{H}_2\text{S}$, ΣPO_4 or ΣNH_4 is used to describe this (or any) reaction does not impact the generated TA. In their response, the authors have thoroughly and correctly shown that the $\Delta\text{TA}/\Delta\text{DIC}$ ratio is 1.14 irrespective of the chosen formulation for the acid-base speciation. In fact, the chosen formulation for the acid-base speciation should also not impact the net generated number of protons, as is nicely shown by Hofmann et al. (Mar Chem, 2010). I also think that the authors have generally responded well to the other comments raised by myself and Reviewer #3. Moreover, I believe that the additional space allowed by Nature Communications improves both the flow and clarity of the manuscript, and that therefore this journal is a more suitable platform for this manuscript. Given the relatively few changes between the current and previous versions, and my overall positive feeling on the current version of this manuscript, I only have some minor comments, which are given below:

We agree, and again we value very highly your inputs over the several rounds of reviews.

Abstract: Here I have a preference for certain formulations used in the previous version of the manuscript, e.g. on line 22 I would replace "lead" with "may lead" and in line 24 I would replace "It is not clear" with "It is less clear".

Agreed and suggestions accepted.

Lines 42-44: I would suggest using either "...contribute to far greater acidification..." or "cause a far greater acidification effect" (or equivalent) here.

I agree and accept the suggestion. The word "effect" is deleted.

Line 106: shouldn't this be August 9, 2013?

Yes, corrected. Thanks.

Lines 135-137: I'm not sure if this was added because of an earlier reviewer comment, but for clarity I would suggest choosing one formulation for Eq. (1) only (possibly the one using H_2S) and removing this sentence.

I agree and accept the suggestion. (Yes, this was added because of a previous review comment).

Line 156: It may be better to stick to $-\Delta\text{TA}$ only here. In terms of stoichiometry, how much H^+ is generated exactly through these processes depends on the ambient pH, i.e. the relative fractions of NH_4^+ versus NH_3 and H_2S versus HS^- (see e.g. again the Hofmann et al. Mar Chem 2010 work).

I agree and change H⁺ to “acid” (which is a total amount of TA reduction) to avoid possible confusion between the exact number of H⁺ production and TA reduction.

Section “Vulnerability of estuarine waters to acidification”: I miss a sentence in this discussion that I considered rather clarifying in the previous manuscript version: “However, our simulations and those of the previous studies suggest that lower buffering capacity itself does not necessarily lead to low pH (Fig. 5a); rather, it allows a much greater pH decrease when other sources of CO₂ or strong acids are added (Fig. 5b).” Any particular reason why this has been removed?

I actually would very much prefer to keep this sentence but somehow one earlier comment made me believe that we need to shorten this section and probably do not need to repeat a “cliché” like this line. But I agree it is an important one worthy of repeating in this context, so I now add it back. Thank you for pointing this out to me.

Lines 405-413: Am I understanding correctly that a range of DIC increases is taken here from both 2013 and 2014 data, whereas the initial conditions of the calculation are from the 2013 data only? Why aren't the pH changes for 2013 and 2014 calculated entirely independently, i.e. by using the measured initial conditions for 2014 and then subtract the 2014 DIC increase? And how is the pH change for April 2015 calculated? Shouldn't, in this case, the DIC change be added to the measured 2015 spring data?

Yes, the range of fluxes are taken from the three cruises (4.3 – 6.7 mmol/m²/d) to calculate the DIC increases from these fluxes. Then they are added to the initial condition in 2013 only (back calculated from summer 2013). The purpose of this calculation is to measure the potential importance of local CO₂ uptake on bottom water pH. Since we do not trace it and compare it with observation (which is dominated by respiration), therefore, I do not feel it is needed to estimate this for 2014 condition as well (but the result won't be much different as DIC and TA in 2014 are not greatly different from those of 2013). So now in Table 1, we deleted the delta-pH calculation for April 2015 as the calculation is meant to see how the accumulated CO₂ flux may influence the summertime bottom water pH.

Lines 504-505: “Given the low H₂S in the bay, in- or excluding H₂S resulted in a minor difference only”
Agreed and changes made.

Fig. 3b: shouldn't the same symbols be used here as in Fig. 3a? (i.e. a distinction between the various data sources). Also, what do the bars in the upper right corner represent, and how have they been defined? If they are “end-members” like in Fig 3a, why are they bars instead of dots?

In Fig. 4a, we showed various locations of the data inside the bay to give readers a sense of spatial patterns of DIC and TA in addition to salinity dependent pattern. The focus of Fig. 4b is to show that the buffer factors are all low inside the bay as compared with that in the offshore endmember and with Gulf of Mexico offshore waters. Therefore for clarity data from various locations are lumped together using only one symbol. The bar in the upper right corner are now replaced with real endmember data.

Reviewer #2 (Remarks to the Author):

In this revised manuscript, the authors investigate how interplay of natural and anthropogenic CO₂

sources, redox reactions, and sediment-water exchanges govern ocean acidification rates in the Chesapeake Bay where eutrophication, hypoxia and anoxia, and low pH occur due to anthropogenic nutrient inputs. The authors carry out in-depth analyses of good quality measurement data for oxygen (O₂), hydrogen sulfide (H₂S), pH, dissolved inorganic carbon (DIC) and total alkalinity (TA). They focus on a pH minimum observed in intermediate depths and argue oxidation of reduced chemicals in bottom water, carried upwards by mixing, is responsible for it. They provide supporting evidence, in the form of subsurface enrichment of TA and DIC for which they invoke sulfate reduction and calcium carbonate dissolution to explain. In order to further substantiate this, they show that simple numerical model incorporating the above processes together with aerobic respiration of organic matter is able to simulate the observed DIC and TA relationship in subsurface water. In conclusion, the authors suggest that synergistic effect from river-ocean mixing, global and local CO₂ uptake from the atmosphere, CO₂ and acid production from respiration, and other redox reactions lead to a poor acid buffering capacity, severe acidification and carbonate mineral dissolution in the Chesapeake Bay and likely other estuaries worldwide.

This is well written manuscript that advances our understanding of the complex processes governing OA rates in coastal ocean. To my knowledge, it is the first of its kind and represents an important contribution to coastal biogeochemistry. The authors have also addressed the earlier reviewer comments adequately. Therefore, I recommend publication in Nature Communications after minor revision in which the authors address my following specific comments.

Thank you.

Page 4, line 92: “are not insignificant” please change to “are significant”

Agreed and the suggestion is accepted.

Page 6, line 126-127: “..and bottom-water pH and TA were lower in 2014 than 2013 (Fig. 2a-e)” To me it seems there is a pH decrease that cannot be explained by DIC and TA. From Figs 2b and 2c it seems that bottom water DIC and TA are lower in 2014 by the same magnitude as compared to 2013 (the x-axis scale is very coarse and I might be wrong here). If so, and if S difference is not significant, why then bottom water pH_{25C} has decreased so much in 2014?

In this paper, most description and all the modeling are based on the August 2013 data. The purpose of presenting August 2014 vertical profile data is to support that the mid-depth pH minimum is not a onetime feature. Therefore we didn't do a quantitative assessment of internal consistency for the 2014 data, which we did for the 2013 data (see Supplementary Fig. 10). As shown in this figure, adding HgCl₂ may lead to a slight inaccuracy in TA data in bottom waters depending on the concentrations of H₂S and possibly the time of sample storage (within 4 weeks).

However a quick plot of DIC:TA to pH for both years (see Fig. R1 below) suggests that there is no substantial difference in the internal consistency between the two years. Also from my excel file, the bottom water TA in 2014 is lower than that of 2013 by 30 umol/kg while the DIC difference is 15 umol/kg. Thus the pH should be lower in 2014. At this stage we do not know why bottom water pH is lower in 2014 than 2013. The [H₂S] is slightly higher in 2014 than 2013 but data are not as complete in 2014 as in 2013. At this stage, I feel it is not productive trying to conclude one way or the other on this

issue. However the lack of this information doesn't affect the conclusions and discussion presented in this paper—when H₂S is mixed upward, acid is produced and the main geochemical processes dominate the bottom water.

Since the 2014 data are for supportive purpose we feel it is best to be presented in the supplement (as the 2014 pCO₂ data). Thus we moved the 2014 data to the NEW supplementary Fig. 6.

Fig R1. DIC:TA vs pH plot on measured carbonate system data.

As a follow up, in 2016, my lab conducted a very detailed field study of 5 cruises in the bay with TA measured on site within hours of collection and without adding HgCl₂ for preservation. As part of a Ph.D. dissertation, I hope a student will report these new results and make comparison with the 2013 and 2014 data in follow-up publications.

Page 15, line 334: can the authors provide any information regarding if they controlled the accuracy of the O₂ sensor?

While salinity, temperature, and O₂ were obtained from the CTD Rosette system using only factory calibration, O₂ data were also compared with those measured by the voltammetry method which was calibrated against Winkler titration as described in the paper by the Luther group for the 2013 cruise reported (ref #36, Oldham et al. 2015).

Pages 15, 340-341: how long were the TA and DIC samples stored before analyses?

Samples were stored in a walk-in cold room (or in the lab refrigerator during the few days before analysis) and analyzed “within 4 weeks” of the cruises. This info is now added to the paper (In Methods).

Page 16, lines 355 – 356: The use of HgCl₂ for DIC/TA sample preservation is quite normal. It is therefore of general interest that the authors report a TA decrease “caused by HgS precipitation, which releases H⁺ when HgCl₂ was used to stop microbial activity...” Is this a general “warning” about TA decrease whenever HgCl₂ is used for preservation? Or do they believe this was specific to their situation?

Yes, the use of HgCl₂ for DIC/TA sample preservation is a quite normal practice. This is not a problem in most coastal waters but will become one when [H₂S] appears in bottom waters. Since Cai lab always over-constrain the carbonate system, a calculation of TA from pH and DIC indicates this problem (supplementary Fig. 10a & b). Fortunately this issue is not serious in the Chesapeake Bay where [H₂S] is not very high. In our follow-up cruises in 2016, we measured TA within hours of sampling without adding HgCl₂. We will report and assess this issue in a more technical paper as part of a student dissertation research.

Finally, I notice that equations are numbered as 1,2,..both in the main text and in the Methods. For unique identification of equations, it may be wise to number them differently. For instance, M1, M2, ..in Methods.

Yes, the name of equation 5 appeared twice in both the main article and methods. now corrected, thanks.

Reviewer #3 (Remarks to the Author):

Recommendation: Accept with minor revisions

I liked the paper and I recommend publication. I have a few general comments the authors should consider along with a few minor clarification points.

The paper presents a novel study that looks at the role of eutrophication and redox reactions on the acidification of Chesapeake Bay. The observations show the existence of a mid-depth pH minimum associated with the oxidation of reduced chemicals - which is both novel and interesting. Further, they show in the anoxic bottom water there is rapid increases in TA and DIC related to SR and CD. These two observed features are linked because the SR reduction produces H₂S which is mixed up into the mid water and contributes to the observed pH minimum. I note while the buffer capacity of the Chesapeake Bay water appears low, the CD in the highly acidic bottom water introduces another way the bay can buffer future rising CO₂ levels and this is an important feature to consider in future OA projections for this Bay. I would like the authors to more clearly acknowledge this mechanism in the revising their conclusion about the future OA projections for the Bay.

Thank you for this suggestion. We certainly agree with your point. We had some debates among coauthors how extensively we should discuss this issue with necessary evidence. Because of the spatial limitation of our previous submission to Nat Geosci, we only pointed this out but didn't emphasize it at all. We now expand a bit on this point ("and thus provides an important buffer mechanism in bottom waters") in line 243 and brought in previous studies (lines 253-257) to support the statement (in p.11). As suggested by the reviewer we also more clearly acknowledge this mechanism in the concluding paragraph of the paper (in lines 328-332).

My other suggestion is to redraw figure 4 to remove the salinity mixing trend to produce a TA-DIC figure that is only influenced by BGC processes. I note that SR and CD appear to have a similar slope in figure 4, is this correct because it appears graphically in the figure that it would be difficult to separate these two processes. The amount of CD seems large does this reflect a large amount of CaCO₃ in the sediments or the dissolution of recently produced CaCO₃ from the upper ocean? If it is the former process, how long before one runs out of CaCO₃ to buffer the OA impact (is this relevant to the OA projections for the Bay?)

First, in Fig. 4a, while the real data always have a mixing component there, most mixing occurs between the surface water and bottom water (there is not much mixing inside the bottom water). Second, For the model simulations, within the S=20 bottom water, there is no mixing. Our plot is exactly a plot of TA vs DIC that is only influenced by biogeochemical processes as suggested by the reviewer. (thus no action is taken)

Third, while graphically the slopes of SR and CD may appear similar (they are not), numerically they are very different (SR slope is 1.14 and CD slope is 2.0). We believe when one reads Fig. 4 together with Table 2, this point is clear.

Fourth, the way Fig. 4a and b are arranged, we present the same biogeochemical processes in both TA vs DIC and pH vs O₂ spaces. This serves a good purpose to show how various processes (and which one at what stage) control bottom water pH.

Fifth, the second part of the comment are very good research questions. Unfortunately we cannot fully answer them in the current paper. We have suggested that recently produced CaCO₃ from the high pH surface is probably responsible for the CaCO₃ dissolution in bottom water though sediment dissolution is another possibility. While measurements of Ca²⁺ and DIC-C13 during our 2016 fieldwork will help to answer these questions further, additional benthic flux study and numerical modeling are needed in future studies. We are expanding a bit on this discussion (also see response to comment below marked as L297).

I'm satisfied with their calculation of TA and DIC changes associated with SR, which was raised in the previous review.

Thank you.

Minor comments

I75, in the mid-bay state how much below atmospheric CO₂ the surface water is

This is later presented in Table 1. We now cite Table 1 right here too.

I116, 6m depth still within the upper mixed layer is this correct? seems inconsistent with a large drop in pH

yes this is correct. Because of the slow kinetics of H₂S oxidation, the acid generation is not limited to the oxic-anoxic boundary but also is extended to above it into the very near surface water. This has been shown more clearly by O₂ and pH microelectrodes profiles and by numerical models in less dynamic sediments (refs cited in the paper). While it is hard to prove it with hard evidence in very dynamic water column, we suggested the same mechanism is operating here too. In future studies, bundled H₂S and pH sensors should be deployed to study this dynamic coupling in situ.

I also believe that chemical gradients can exist in seemingly well mixed water (indicated with no salinity gradient). For example, during wintertime in Gulf of Mexico, we often see DIC and pH differences between surface and bottom waters in areas of only very small salinity gradient. In Fig. 2 while salinity gradient is rather small in the surface 5-6 m, O₂ and pH gradients are quite clear.

I212, change CB to CD

Corrected. Thanks for spotting this.

I273, the poor buffering capacity of Chesapeake Bay water is interesting but in the bottom water of the bay it is the dissolution of CaCO₃ that partially buffers the increase in CO₂ (external source for buffering). (yes, we agree) Does the low buffering capacity of the water contribute to the low CaCO₃ saturation state (yes, exactly, see Table 2) and do this set up the ideal conditions for high CaCO₃ dissolution? (yes, we think so as $\omega(\text{arag})$ is very low, < 0.4)

I297, and in the conclusion, Yes! as you already showed these processes would have a significant impact on TA and DIC values. Would this substantially alter the sensitivity of the bay to future OA? it appears the Bay has a natural way to buffer OA by dissolving CaCO₃, you should discuss this if you want to comment on future OA response of the bay.

As said earlier these are excellent research questions that need be answered in future research. It is also an excellent suggestion that I “should discuss this if [we] want to comment on future OA response of the bay”. Thus we expand the paragraph discussing CaCO₃ dissolution (p.11) and add a bit of this in the concluding paragraph (lines 328-332).

Figure 4, not clear if both the sequential and simultaneous lines start at the same initial point? Do they? if not why not?

Yes, both the sequential and simultaneous lines start at the same initial point. When both [O₂] and Omega are low, the sequential simulation is allowed to start (this is explained in the Methods) and can be seen in Fig. 4b when the aerobic respiration line separate into two lines at [O₂] about 80 $\mu\text{mol/kg}$.

Figure 5. state that panel c lines come from the difference between the lines shown in panels a and b. okay, suggestion accepted. Thanks.

Reviewers' Comments:

Reviewer #1:

Remarks to the Author:

To me, the authors have addressed my comments on the previous manuscript version, as well as those of the other reviewers, in a satisfactory way. In contrast to my previous revisions, I have therefore not read the revised manuscript in great detail at this stage.

In terms of the reviewers' comments and author responses, I have only one comment left: I am totally fine with moving the 2014 data to the supplement, but I would in this case suggest to also remove the 2014 data from Fig 4b (and Fig 4a, although I am not sure if they are presented there), especially since the model simulations only concern the 2013 data and presenting the data from both years does not aid the interpretation of this figure.

The additional discussion on carbonate dissolution has improved the manuscript and I look forward to seeing results of the detailed field study of 2016 in future publications, especially on the sedimentary processes. In summary, I would recommend publication of the manuscript.

Reviewer #2:

Remarks to the Author:

The authors have satisfactorily addressed the points I raised, and I have only one minor further suggestion. For complicity, please modify lines 130-131 to "..., and bottom-water pH, DIC, and TA were lower in 2014 than 2013 (Supplementary Fig. 6)."

finally, I take this opportunity to congratulate the authors with a nice work.

Reviewer #3:

Remarks to the Author:

I'm satisfied with the authors responses to the reviews so I'm happy to recommend accepting the paper for publication.

Point-by-point response to referee comments

REVIEWERS' COMMENTS:

Reviewer #1 (Remarks to the Author):

To me, the authors have addressed my comments on the previous manuscript version, as well as those of the other reviewers, in a satisfactory way. In contrast to my previous revisions, I have therefore not read the revised manuscript in great detail at this stage.

In terms of the reviewers' comments and author responses, I have only one comment left: I am totally fine with moving the 2014 data to the supplement, but I would in this case suggest to also remove the 2014 data from Fig 4b (and Fig 4a, although I am not sure if they are presented there), especially since the model simulations only concern the 2013 data and presenting the data from both years does not aid the interpretation of this figure.

<response: this is exactly what I did/do. The model simulation is based on the 2013 data only. I now added in the Fig. 4 caption "...from August 2013" to make this clear.>

The additional discussion on carbonate dissolution has improved the manuscript and I look forward to seeing results of the detailed field study of 2016 in future publications, especially on the sedimentary processes. In summary, I would recommend publication of the manuscript.

<response: thanks>

Reviewer #2 (Remarks to the Author):

The authors have satisfactorily addressed the points I raised, and I have only one minor further suggestion. For complicity, please modify lines 130-131 to "...and bottom-water pH, DIC, and TA were lower in 2014 than 2013 (Supplementary Fig. 6)."

<response: done>

finally, I take this opportunity to congratulate the authors with a nice work.

<response: thanks>

Reviewer #3 (Remarks to the Author):

I'm satisfied with the authors responses to the reviews so I'm happy to recommend accepting the paper for publication.

<response: thanks>